# Performance Evaluation of Real-Time Kinematic Global Navigation Satellite System with Survey-Grade Receivers and Short Observation Times in Forested Areas

**DOI:** 10.3390/s24196404

**Published:** 2024-10-02

**Authors:** Mihnea Cățeanu, Maria Alexandra Moroianu

**Affiliations:** Department of Forest Engineering, Forest Management Planning and Terrestrial Measurements, Faculty of Silviculture and Forest Engineering, Transilvania University of Brașov, Șirul Ludwig van Beethoven 1, 500123 Brașov, Romania

**Keywords:** RTK-GNSS, accuracy and precision of GNSS observations, forestry

## Abstract

The Real-Time Kinematic (RTK) method is currently the most widely used method for positioning using Global Navigation Satellite Systems (GNSSs) due to its accuracy, efficiency and ease of use. In forestry, position is a critical factor for numerous applications, with GNSS currently being the preferred solution for obtaining such data. However, the decreased performance of GNSS observations in challenging environments, such as under the forest canopy, must be considered. This paper analyzes the performance of a survey-grade GNSS receiver under coniferous/deciduous tree cover. Unlike most previous research concerning this topic, the focus here is on employing a methodology that is as close as possible to real working conditions in the field of forestry. To achieve this, short observation times of 30 s were used, with corrections received directly in the field from a Continuously Operating Reference Station (CORS) of the national RTK network in Romania. In total, 84 test points were determined, randomly distributed under the canopy, with reference data collected by topographical surveys using total station equipment. In terms of the overall horizontal accuracy, an RMSE of 2.03 m and MAE of 1.63 m are found. Meanwhile, the overall vertical accuracy is lower, as expected, with an RMSE of 4.85 m and MAE of 4.01 m. The variation in GNSS performance under the different forest compositions was found to be statistically significant, while GNSS-specific factors such as DOP values only influenced the precision and not the accuracy of observations. We established that this methodology offers sufficient accuracy, which is application-dependent, even if the majority of GNSS solutions were code-based, rather than carrier-phase-based, due to strong interference from the vegetation.

## 1. Introduction

Since their introduction in the 1980s, Global Navigation Satellite Systems (GNSSs) have brought about a paradigm shift in all fields that are dependent on position [1]. One such area is the management of natural resources, where satellite positioning has become an invaluable tool for administrators [2]. The integration of satellite positioning in forestry operations has been somewhat slower compared to other areas, mainly due to the difficulties of obtaining accurate data under forest canopies [3], as the reduced performance of GNSS receivers in forested environments has been an issue of interest in recent decades [4,5]. However, the continuous improvement of the technology has alleviated these concerns and led to GNSSs becoming pervasive in forest management [6] and an essential component of precision forestry [7]. As examples of the various uses of satellite positioning in forestry, we can note the following: the positioning of sample plots for forest inventory to facilitate revisitation [4], the mapping of ownership or management unit boundaries [8], the planning and execution of forest roads [7,9], the monitoring of harvesting operations [10,11], the integration of field measurements with remotely sensed data [12] or the tracking and monitoring of wildlife [13]. Evidently, applications related to these operations are characterized by different requirements in terms of positioning accuracy.

Forest cover will prevent some of the radio signals from reaching the GNSS antenna on the ground and therefore lead to either a degradation of accuracy [5] or even the impossibility of obtaining location data. Therefore, it can be argued that the most important factor leading to the adoption of satellite positioning in forestry research and application is the constantly increasing number of GNSS satellites in operation. Presently, the four GNSS constellations (NAVSTAR GPS, GLONASS, Galileo and BeiDou) amount to more than 100 satellites orbiting the Earth. Other important advancements that took place in recent decades are (1) the introduction of additional satellite signals that allow dual- or triple-frequency receivers to model the ionospheric error [14], (2) the evolution of differential GNSS methods, such as the Real-Time Kinematic (RTK) method, that make use of corrections determined over known points (bases) to improve the positioning accuracy [15], (3) the expansion of regional, national or even continental RTK networks [16] that allow users access to differential GNNSs without the need of setting up their own bases and (4) the implementation of auxiliary systems meant to increase the accuracy and reliability of satellite positioning, such as the Wide Area Augmentation System (WAAS) in North America [17] or the European Geostationary Navigation Overlay Service (EGNOS) [18]. In addition, we must also note the continuous improvement of GNSS receivers in terms of hardware and software capabilities. 

NAVSTAR GPS, the initial GNSS system, was developed with the intention of providing position data in open spaces [19]. Currently, especially due to the proliferation of the RTK method, which is still the most accurate method and can achieve (sub)centimeter-level errors [20], we can consider the issue of accuracy to be solved in the case of open space for all but the most highly demanding applications (in terms of positioning accuracy). This is not the case for challenging environments, such as urban canyons, dense forest canopies or steep slopes, all of which have a significant impact on GNSS satellite signals [21] and can therefore cause severe degradations in terms of accuracy and reliability [22]. There are two main issues associated with such environments: (1) a significant reduction in the number of line-of-sight (LOS) satellites due to the presence of obstructions [23] and (2) a higher incidence of multipath occurence, when radio signals do not reach the receiver antenna directly but after being reflected by buildings, branches, tree trunks or the ground surface. Unlike most other error sources, the effect of multipath occurence cannot be reduced through differential techniques [24] and is presently considered the most impactful error source [25], especially in challenging environments such as under forest canopies [26,27].

Given the rapid changes in GNSS and GNSS-adjacent technologies, their potential has to be constantly re-evaluated [6], especially in conditions that cause signal obstructions where there is still a potential for improving the accuracy. Because of this, many studies undertaken over the last few decades have focused on satellite positioning in forested terrain. The reported accuracies vary wildly, which is to be expected given the vast differences in terms of instruments, the number of satellite constellations in operation at the time of publication, the use of RTK vs. static determinations, the length of recording sessions and forest conditions [11]. Therefore, the following literature review will mostly focus on the qualitative findings of previous research and less so on reported numerical values of accuracy indicators. One of the first comprehensive studies of GPS accuracy under the forest canopy focused on the influence of the antenna height [4]. The study reported a better signal reception (and therefore improved accuracy) at antenna heights of 4 m compared to heights of 2 m and a linear relationship between errors and the Point Dilution of Precision (PDOP). The authors of [28] carried out an extensive survey campaign using differential GPS, collecting data over nine months for 27 sites located inside the forest and concluding that the forest type is the most important factor influencing accuracy, with mean errors of 8.4–8.5 m in hardwood stands and 7.8–11.3 m in conifers, dependent on the terrain. The authors of [5] report a mean horizontal error of approx. 1 m for observation times of 10 min, showing that the canopy density, tree species and observation times influence accuracy. In [29], the authors tested the performance of GPS+GLONASS in open vs. forested areas, reporting centimeter-level accuracies but with long observation times (300–350 min per point) and highlighting the difficulty of GNSS signal reception under the canopy. The authors of [30] used a survey-grade receiver and differential GNSS to position tree stems in various canopy conditions, reporting more accurate positions in conifers and horizontal RMSEs in the range of 14–22 m. Ref. [26] focused on individual tree characteristics, rather than general forest conditions, establishing that the location of nearby trees, rather than their size, is the primary factor affecting horizontal accuracy. Ref. [6] also reported a possible interaction between individual tree locations and accuracy but without focusing on this matter. A degree of influence from forest conditions on GNSS performance is commonly reported [6,26,31,32], but in some cases, the correlations are not found to be statistically significant [15,33], or trends are characterized as not consistent [26]. In other cases, no significant influence of forest composition on GNSS performance is found [3,34,35].

Attempts at improving the accuracy of positioning in forest conditions have been made, with varying degrees of success. The most common ones involve the use of dual-frequency receivers [1,9], differential GNSS [5,15,25] or Precise Point Positioning (PPP) [15], signals from multiple constellations to increase the number of visible satellites [29,36], a system composed of three receivers mounted on a metal beam [25,37], different methods of holding the receiver [3], the addition of a metal plate under the receiver to reduce multipath [1] or the fusion of GNSS determinations with data provided by an Inertial Measurement Unit (IMU) [11]. 

Given the rapid evolution of satellite positioning methods and the lack of consensus regarding the interaction of environmental variables (especially forest composition) and GNSS accuracy, this study aims to analyze the accuracy and precision of a survey-grade receiver operating in RTK mode for fast (30 s) point determinations in two test plots (with deciduous and coniferous vegetation cover, respectively). The short observation times used here are a consequence of focusing on practical uses of GNSS receivers in field data collection campaigns carried out for forestry applications. This also permits a larger sample size compared to most other previous research. Both the horizontal and vertical accuracy and precision were considered, along with factors such as the forest type, ground slope, canopy density as estimated by the Normalized Difference Vegetation Index (NDVI), DOPs or the number of visible satellites. All tree stems positions were determined using traditional, high-accuracy surveys with total station equipment to also account for the number of and distances to trees around GNSS-determined points.

## 2. Materials and Methods

### 2.1. Study Area

This research was carried out in a forested area located in the central part of Romania (45°43′30″ N, 25°39′19″ E), part of the Natura 2000 site “Lempeș Citadel Hill–Hărman swamp” (Figure 1). The area is almost completely covered by dense forest vegetation and is characterized by a hilly terrain, with relatively steep slopes and an elevation range of 500–700 m. Two test plots were established in this area, one inside a Scots pine (*Pinus sylvestris*) forest stand and one inside a deciduous stand composed mainly of beech (*Fagus sylvatica*) and oak (*Quercus robur*). The pine plot has an area of approx. 11,000 m^2^, while the deciduous one has an area of approx. 25,000 m^2^. General forest characteristics for the management units in which the plots are located, as published in the latest forest management plan, are presented in Table 1.

### 2.2. Field Data Collection

To estimate the accuracy of GNSS positioning in forest environments, points were distributed over the test plots and marked with wooden stakes. While points were for the most part randomly chosen, care was taken not to position them too closely (<0.5 m) to tree trunks, which would hinder the operation of the GNSS receiver. Data were collected using a S10 survey-grade GNSS receiver (Stonex S.r.l., Milan, Italy), which can receive satellite signals from all four global constellations that are presently in use (NAVSTAR GPS, GLONASS, Galileo and BeiDou) and has a stated accuracy for RTK operation of 2.5 mm + 0.1 ppm RMS (horizontal) and 3.5 mm + 0.4 ppm RMS (vertical). Determinations were made using the RTK method, with the receiver obtaining corrections directly in the field from a Continuously Operating Reference Station (CORS) of the national RTK network in Romania (ROMPOS—Romanian Positioning System). The receiver was mounted over each point using a monopod with leveling (Figure 2a), and 30 positions (1 s epochs) were determined to obtain average positions. The receiver identified the SFGH reference station as being closest, at approx. 19 km from the study area, and therefore used it for correcting the obtained coordinates. A mask angle of 10° was used to filter out signals from satellites close to the horizon in order to reduce signal interference and the chance of multipath occurring [7]. The antenna was kept at a constant height of 2.50 m. This ensures that the position of the operator next to the receiver does not influence determinations, while also reducing the accidental antenna movements due to wind. It must also be noted that the purpose of this study was not to obtain the most accurate positions possible, but rather to obtain any positions and then determine their accuracy. To achieve this, a high threshold for PDOP of 8 was used, and the receiver recorded data as long as it was in connection with the reference station, even if it did not have a Fixed/Float solution. For this same reason, no SNR (Signal-to-Noise Ratio) thresholds were used. In other words, the GNSS data collection was interrupted only by the occasional, short-term loss of GPRS signal which caused the receiver to enter autonomous mode. WGS84 coordinates specific to GNSS observations were transformed into planimetric coordinates (X, Y, Z), using the official grid coordinate system in Romania (Stereographic 1970, EPSG: 31700) and the TransDatRO software v. 4.08 developed by the Romanian National Center of Cartography, which is integrated directly in the receiver’s operating system.

The coordinates of points were also determined by traditional ground surveys carried out with the total station equipment to serve as a reference (ground truth) against which to compare GNSS positions. For the beech plot, a closed-loop traverse was surveyed, with 5 points inside the canopy and 3 points outside it. These 3 points were determined as references for traverse calculation, using RTK-GNSS and the same SFGH reference station. Each point was determined twice, once in the morning and once in the evening, to ensure significant changes in satellite geometries (Figure 2b). For each observation, a mean position was calculated based on 300 1-sec epochs (about 5 min of observation time). In this case, a PDOP threshold of 2 was used, and only Fixed solutions were stored. The “cold start” method was used for these determinations, meaning that the GNSS receiver was powered off and restarted after each set of 300 epochs. The two pairs of coordinates obtained for each reference point were compared, found sufficiently close to each other for all points (Table 2), and averaged to obtain the final coordinates. In a similar manner, for the pine plot, another closed-loop traverse consisting of 7 points was determined, with 3 points located outside the canopy and used as reference. Traverse misclosures were in the range of 2–4 cm (horizontal) and 6–8 cm (vertical). During the topographical survey, the location of tree trunks was also determined, with 1185 trees identified (432 in the coniferous test plot and 753 in the deciduous plot, respectively).

In total, 89 test points were determined, 65 in the deciduous and 24 in the pine plot. The relatively low number of points in the pine stand is due to the extensive presence of under-canopy vegetation, which hindered both operator movement and visibility from the total station’s location. Field data collection was conducted in February–March 2024, so the deciduous stand was in leaf-off condition. The canopy conditions of the coniferous test plot are presented in Figure 3, while Figure 4 shows the conditions of the deciduous test plot.

### 2.3. Accuracy and Precision Estimates

Repeated GNSS observations (30 per point in this study) provide an estimate of precision, but the accuracy must be established by comparison with external data as reference (in this case, coordinates obtained by traditional surveying techniques). The precision is calculated and stored directly by the GNSS receiver for the horizontal and vertical directions and is simply the standard deviation of repeated observations recorded for each point.

Regarding accuracy, this was also considered for the horizontal and vertical components and involves the calculation of horizontal (*e_H_*, which will also be referred to as displacement) and vertical errors (*e_Z_*) [32]:(1)$$e_{H}=\sqrt{{\left(X_{GNSS}-X_{Truth}\right)}^{2}+{\left(Y_{GNSS}-Y_{Truth}\right)}^{2}}$$
(2)$$e_{H}=\sqrt{{\left(X_{GNSS}-X_{Truth}\right)}^{2}+{\left(Y_{GNSS}-Y_{Truth}\right)}^{2}}$$
(3)$$e_{Z}=Z_{GNSS}-Z_{Truth}$$
where (*X_GNSS_*, *Y_GNSS_*, *Z_GNSS_*) are the coordinates determined by the receiver, and (*X_Truth_*, *Y_Truth_*, *Z_Truth_*) are reference coordinates from the topographical survey.

Once individual errors are known, the accuracy can be determined for groups of points, stratified based on the forest type (i.e., test plot) or for categories of factors such as the number of Float solutions, the number of satellites and so on. In this case, accuracy is estimated by the Mean Unsigned Error (also called Bias), Mean Absolute Error (MAE) and Root Mean Square Error (RMSE) [8]:(4)$$Bias=\frac{\sum_{i=1}^{n}e_{i}}{n}$$
(5)$$MAE=\frac{\sum_{i=1}^{n}\left|e_{i}\right|}{n}$$
(6)$$RMSE=\sqrt{\frac{\sum_{i=1}^{n}e_{i}^{2}}{n}}$$
where *e_i_* is the error (horizontal or vertical) corresponding to the *i*-th point out of the *n* total points included in the respective group. Note that since horizontal errors represent displacements and are always positive, Bias is only reported for vertical errors.

### 2.4. Potential Factors Considered as an Influence on GNSS Accuracy and Precision

Several factors were considered for the analysis of GNSS accuracy in forested areas. These factors can be classified thus:GNSS factors: Point Dilution of Precision (PDOP), Horizontal Dilution of Precision (HDOP), Vertical Dilution of Precision (VDOP), number of visible satellites and the proportion of Float/DGPS solutions.Environmental factors: ground slope (calculated from a DTM interpolated at a 5 m resolution from *Z_Truth_* values), forest composition (which test plot the points belong to) and NDVI (calculated in Google Earth Engine at a 20 m resolution from cloud-free Sentinel-2 imagery collected between 2022 and 2024).Factors related to tree locations: distance to the nearest tree, number of trees within radii of 2, 4, 6, 8 and 10 m around each point and the average distance to trees in these radii.

## 3. Results

### 3.1. GNSS Accuracy in Coniferous/Deciduous Forest Conditions

The statistical indicators of accuracy were calculated for all 89 surveyed points and stratified by forest type (Table 3). The accuracy, especially in terms of horizontal displacement, is relatively good given the challenging conditions, with an MAE of 1.63 m and an RMSE of 2.03 m. The most accurate horizontal position corresponds to a beech point, which had a horizontal displacement of 0.15 m. Meanwhile, the most inaccurate position is also found in the beech/oak stand and has a displacement of 6.74 m. It is apparent that the horizontal accuracy is improved in the beech/oak stand, with the RMSE increasing by 34% (from 1.84 to 2.47 m) and median error increasing by 71% (from 1.21 to 2.07 m) in the pine stand. 

In terms of the vertical accuracy, while 3 of the 89 points exhibit negative vertical errors (ranging from −0.02 to −1.462), it is evident that the GNSS observations tended to overestimate the elevation (with Bias values of 4.40 and 2.76 m for the beech/oak and pine points, respectively). Also, here we find a reversal of the situation highlighted for horizontal accuracy, with observations in the pine plot having better a vertical accuracy than those in beech/oak, with the latter case showing an increase of 51% for the RMSE (from 3.49 to 5.27 m) and 36% for the median error (from 2.65 to 3.61 m).

In terms of the statistical distribution (Figure 5), while the W values of the Shapiro–Wilk normality tests show that the distribution of error values is close in shape to the normal distribution, the *p*-values indicate that the null hypothesis can be rejected, suggesting that errors significantly deviate from a normal distribution in all cases, with the sole exception of vertical errors in the pine stand (*p*-value = 0.22, *n* = 24). Given the results of the Shapiro–Wilk test, the two samples (coniferous vs. deciduous GNSS observations) were compared at a 0.05 significance level using the Wilcoxon rank-sum test, which does not assume normality [38]. This showed significant differences between the two plots in terms of horizontal (*W* = 562, *p*-value = 0.044) and vertical accuracy (*W* = 1038, *p*-value = 0.017). 

In terms of the direction of displacement, a tendency of points in the pine plot to be shifted towards the west and south is found, while points in the beech/oak plot tended to be shifted towards the east and west/south-west (Figure 6).

### 3.2. Variation in GNSS Conditions under the Forest Canopy

As previously discussed, several factors related to GNSS data collection and recorded by the receiver were considered: PDOP, HDOP, VDOP, the number of visible satellites and the type of GNSS solution (Table 4). Note that no Fixed solutions were obtained, with the 30 GNSS determinations for each point being split between Float and DGPS solutions.

DOPs indicate consistently appropriate satellite configurations in both test plots, with DOP values under 4 for all points (Figure 7). This is due to the high number of visible satellites, which had an average of approx. 13 and a minimum of 8 satellites in both forest stands. The similarity of GNSS conditions for the two test plots is also evidenced by the Wilcoxon rank sum test, with *p*-values between 0.074 and 0.303 for the five factors considered. In terms of solutions, for the most part, only DGPS solutions were achievable under the dense canopy cover, with 83% of points having 30 out of 30 DGPS solutions and only 3 with at least 10 out 30 Float solutions (1 point in pine and 2 in beech/oak, respectively).

### 3.3. Distribution of Trees around Test Points

The distances between points and their nearest tree range from 0.20 to 9.44 m, with an average of 2.76 m. Note that while the point positions were chosen so that no tree was closer than 0.5 m, this was estimated visually in the field, so four of the points (two in each test plot) ended up being closer than 0.5 m to a tree. In terms of the number of trees located around points inside various radii (from 2 to 10 m, in increments of 2 m), we must note that the 2 m radius was excluded from further analysis, as 75% of observations had 0–1 trees inside that radius (Table 5).

### 3.4. Importance of Factor Variation on GNSS Accuracy

External factors were tested in terms of their relative importance in explaining the variation in horizontal/vertical accuracy and precision. For brevity, only the four most important factors are presented (Table 6).

The factors considered in this study do a rather poor job of estimating the overall accuracy (25–27% of variation explained), but this is mostly due to the deciduous test plot. When only points inside the pine stand are considered, we find that the factors involved explain more than 80% of the variation in horizontal/vertical accuracy. On the other hand, the GNSS precision (i.e., the standard deviation of the 30 positions determined for each point) is better correlated with the factors considered, with almost half of the variation explained overall and nearly all variation (86–98%) explained for the coniferous test plot. Some additional findings are also worth highlighting. First, DOPs only influenced precision, having virtually no influence on accuracy. Second, the ground slope appears to only influence vertical accuracy. Third, factors related to tree location (distance to the nearest tree, number of and average distances to trees in various radii) only have somewhat of an influence in the coniferous plot but overall do not seem to be deciding factors on the accuracy or precision.

These findings are supported in part by the results of the Analysis of Variance (ANOVA) tests, which identify several factors as having a significant influence on the GNSS accuracy (*p*-values < 0.05): When all points are considered together: Species is significant for both horizontal and vertical accuracy, while No. of satellites and Slope are significant only for vertical accuracy.For the coniferous plot: No. of trees in a 4 m radius and Distance to nearest tree are significant for horizontal accuracy, while no factors are identified as significant for vertical accuracy.For the deciduous plot: no factors are identified as significant for either horizontal or vertical accuracy.

Regarding the precision of GNSS observations, the ANOVA showed the following:When all points are considered together: PDOP and No. of satellites are significant for horizontal/vertical precision, while Species and Slope are significant only for horizontal and vertical precision, respectively.For the coniferous plot: No. of satellites is significant for horizontal precision, with no significant factors for vertical precision.For the deciduous plot: PDOP is significant for horizontal precision, while Slope is significant for vertical precision.

### 3.5. Level of Agreement between GNSS Accuracy and Precision

The precision of GNSS-determined positions is commonly expressed by the standard deviation of repeated observations of the same point. Therefore, the standard deviations of the GNSS coordinates were compared with the horizontal/vertical accuracy (i.e., the displacement of point locations and vertical errors, respectively). The results show little to no correlation between accuracy and precision, especially for the horizontal component (Figure 8). The overall *r*-value for the horizontal component is 0.09, while for the vertical component, an *r*-value of 0.35 is found. Meanwhile, in pine, *r*-values of 0.06 (horizontal accuracy/precision) and −0.11 (vertical accuracy/precision) are found. On the other hand, the accuracy is somewhat better correlated with precision in the beech/oak plot, with *r*-values of 0.21 (horizontal) and 0.46 (vertical).

## 4. Discussion

### 4.1. GNSS Accuracy in Coniferous/Deciduous Forest Conditions

As previously discussed, in recent decades, the term GNSS has come to cover a wide range of instruments, methods and technologies. This makes quantitative comparisons of accuracy between studies difficult and not particularly relevant, especially when also considering differences in terms of experimental design. As an example, the authors of [29] obtained centimeter-level accuracy using RTK under the forest canopy, but only by allowing the receiver to resolve ambiguities, which led to much longer observation times than in our study (up to 6 h per point). On the other hand, the authors of [30] also used RTK-GNSS in forest conditions and obtained significantly worse accuracies than those reported here (RMSE values of 16–20 m), but this could be explained by the fact that the GNSS receiver was positioned very close to the tree trunks (30 cm) in order to determine their position, which was not the case for this study.

However, by considering the qualitative (rather than quantitative) findings, our results can be put into a larger context. First, while the influence of a canopy presence itself was not a focus of this study, it is well established that RTK-GNSS can easily achieve centimeter-level accuracy in open ground, so our findings (RMSE of 2.03 and MAE of 1.63 m for horizontal accuracy) clearly indicate a significant degradation of GNSS performance in forest environments. This is in line with previous studies [3,9,34,37,39]. As far as vertical errors are concerned, our results show a decrease in accuracy of approx. 2x. Similar proportions have previously been reported [29,36], and expecting a two-fold degradation of accuracy when using GNSS to determine altitude instead of location has been a sort of rule-of-thumb for some time [40]. Secondly, differences between errors depending on forest compositions were found to be statistically significant, which is in line with some [6,33], but not all, previous studies [3,34]. Note that the forest composition (for, e.g., pine vs. beech/oak, in this case) is a necessary simplification that does not account for stem density, differences in age among trees or the presence and type of sub-canopy vegetation, among other conditions that might potentially influence GNSS performance. This could explain the seemingly contradictory findings reported previously. The fact that the horizontal accuracy was better in deciduous forest is explained by the leaf-off conditions in which the GNSS data were collected. However, the relatively better vertical accuracy in the pine plot is not easily explained. This behavior could potentially be attributed to the small sample size for pine (*n* = 24), the difference in slope (pine avg. slope: 19%; beech/oak avg. slope: 24%) and aspect (the pine plot is mostly oriented to the SW, while the beech/oak is oriented towards the NW) or, most likely, a of combination of these factors. Without additional data, these hypotheses cannot be tested.

### 4.2. GNSS Solution Type under the Forest Canopy

GNSS solutions can be of three types, which in order of increasing accuracy are as follows: DGPS (code-based positioning), Float (the receiver has locked onto satellite signals for enough time to allow for more accurate carrier-phase positioning) and Fixed (when enough time has passed so that the receiver can resolve the integer ambiguity that is specific to carrier-phase measurements). In this study, no Fixed solutions were achieved, with almost all epochs recorded as DGPS solutions (97% of 2670 solutions—30 per point). The authors of [37] also report achieving Fixed solutions only for a few percent of observations. This potentially helps explain the relatively low accuracy, especially given the high number of visible satellites (13 on average) and good PDOP values (the average PDOP is under 2). The time-to-fix is outside the scope of this study so was not comprehensively tested, but it should be noted that, at times, the GNSS receiver was left installed on the monopod under the canopy for as long as 1–2 h, and no Fixed solution was achieved (only Float, given sufficient time). The difficulty of carrier-phase determinations under the canopy due to signal interruption hindering integer ambiguity resolution has been established previously [7,9,31,32]. GNSS users operating in forest ecosystems should also take into account that even if a Fixed solution is achieved, the integer resolution could be wrong, which is more likely to happen in challenging conditions [9]. This could lead to the unexpected situation in which Fixed determinations are affected by large errors [23] and are therefore less accurate than Float/DGPS ones.

### 4.3. Relative Importance of Factors for GNSS Accuracy/Precision

In terms of external factors considered, our main finding is that, as far as accuracy is concerned, environmental conditions (such as tree stand species, NDVI or ground slope) tend to explain more in terms of GNSS performance than GNSS-specific indicators such as the number of visible satellites or DOPs. This is to be expected, as the accuracy of satellite positioning is a function of the environment in which determinations are carried out [37]. Ref. [19] also reported that forest structure parameters have a higher influence on satellite positioning than factors related to GNSS, while the authors of [32] showed that the main driver of both accuracy and precision of GNSS are the canopy characteristics. Overall, the factors considered here explain most of the errors found in coniferous conditions (over 85% of variance explained, for both horizontal and vertical accuracy), but this does not apply to the deciduous plot, where only 17–29% of the error variance is explainable. As far as precision is concerned, GNSS-specific factors (especially PDOP, HDOP and VDOP) explain significantly more of the variation in performance for both test plots. The only case in which precision is poorly explained is for the vertical precision in the deciduous plot (29% of variance explained). This is another indicator that the unexpectedly large vertical errors for this plot (significantly higher than the ones in the coniferous plot) are due to unconsidered factors, warranting further study.

Factors related to the distribution of trees around the GNSS receiver rarely showed predictive power, especially in the case of horizontal accuracy/precision. This would indicate that, rather than the tree trunks, factors related to the tree crowns (size, shape and so on) which were not considered for this study would likely have more of an impact on GNSS performance. Ref. [26] report some predictive power for the location of nearby trees but establish that the size of trunks has no correlation to errors, which we believe supports this hypothesis.

### 4.4. On Accuracy vs. Precision of GNSS-Determined Positions

The issue of accuracy/precision is an important one, as in practice, users would rarely have access to reference, ground-truth data to estimate the accuracy of GNSS positioning. Rather, the precision has to be relied upon as a substitute for accuracy. While it is expected that precision underestimates accuracy, given that accuracy is a summation of precision and systematic and gross errors [41], it is desirable that the two are correlated. This would at least imply that precision estimates (given, for example, by the standard deviation of coordinate values for repeated determinations of the same points) allow for comparisons of the relative accuracy achieved. However, our results indicate that precision estimates have very little in common with accuracy, especially for latitude/longitude determinations. Standard deviations of computed coordinates have previously been reported as insufficient for estimating GNSS accuracy [33,42,43]. On a related note, Dilution of Precision (DOP) indicators were found to be better correlated with precision than accuracy. As an example, *r* increases from 0.09 (correlation of PDOP with horizontal errors) to 0.45 (correlation of PDOP with horizontal precision). Previous studies also found PDOP to have little or no predictive power for accuracy [7,42]. With the increase in GNSS satellites in orbit, improper satellite geometries would only be found in the most challenging environments, meaning that currently, the utility of DOPs in explaining GNSS performance is questionable.

## 5. Conclusions

GNSSs benefit from constant technological advances and breakthroughs; therefore, testing their performance is an ongoing concern [44]. The aim of this study was to evaluate the performance of GNSS positioning using a survey-grade receiver for the challenging case of relatively steep terrain covered by dense forest vegetation. The experimental design (short observation times of 30 s/point using RTK with corrections provided by a CORS of Romania’s national system of GNSS reference stations) aimed to mimic, at least in part, the practical usage of GNSS by operators in the forestry sector. Our results show that if using this methodology, meter or even sub-meter accuracy is achievable, even with code-based determinations due to the satellite signal interference due to the presence of vegetation. Horizontal/vertical errors show statistically significant differences between coniferous and deciduous stands. However, more structural forest parameters need to be considered to better explain the variation in GNSS performance. Our analysis highlighted that, under dense forest canopies, environmental conditions might affect the accuracy to a larger extent than GNSS-specific factors such as the number of satellites and PDOP values. Presently, the most critical issue for satellite positioning in forest areas would be the mitigation of multipath, which is an active area of research [24]. 

Overall, using survey-grade receivers and RTK-GNSSs with corrections from national/regional networks allows for rapid positioning under the canopy at adequate accuracies (especially if altitude data are not required), depending of course on application requirements. Further advancements of GNSSs are expected, especially in terms of antenna design, instruments or software processing, which imply that a continuous reassessment of the technology’s potential is warranted [3]. The added benefit of increasing the observation time or the potential of modern, RTK-enabled handheld receivers (such as those integrated into smartphones or even smartwatches) to achieve a similar performance in forest conditions to that of more cumbersome survey-grade receivers are issues to be addressed in further studies.

## Figures and Tables

**Figure 1 sensors-24-06404-f001:**
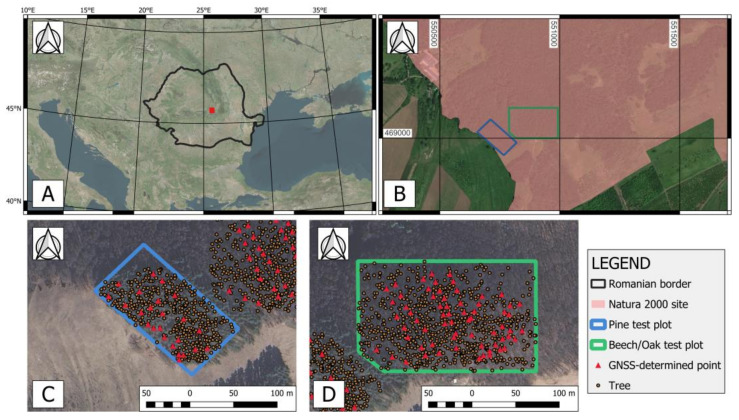
(**A**) General location of study area (red rectangle); (**B**) location of test plots inside the “*Lempeș Citadel Hill–Hărman swamp*” natural reservation; (**C**) pine test plot; (**D**) beech/oak test plot.

**Figure 2 sensors-24-06404-f002:**
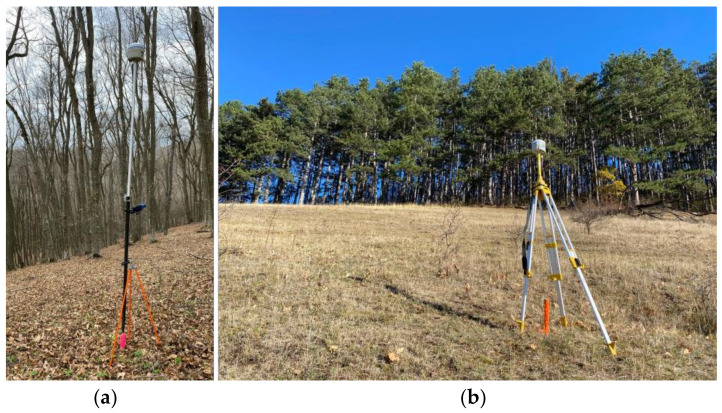
(**a**) GNSS determinations with a Stonex S10 survey-grade receiver for test points; (**b**) GNSS determinations for control points for closed-loop traverse determination.

**Figure 3 sensors-24-06404-f003:**
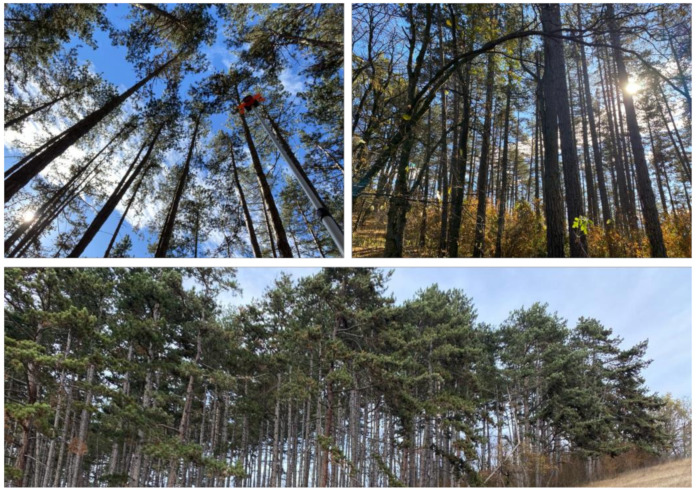
Canopy conditions of the test plot located in a pine stand; note the presence of under-canopy vegetation (**upper right**).

**Figure 4 sensors-24-06404-f004:**
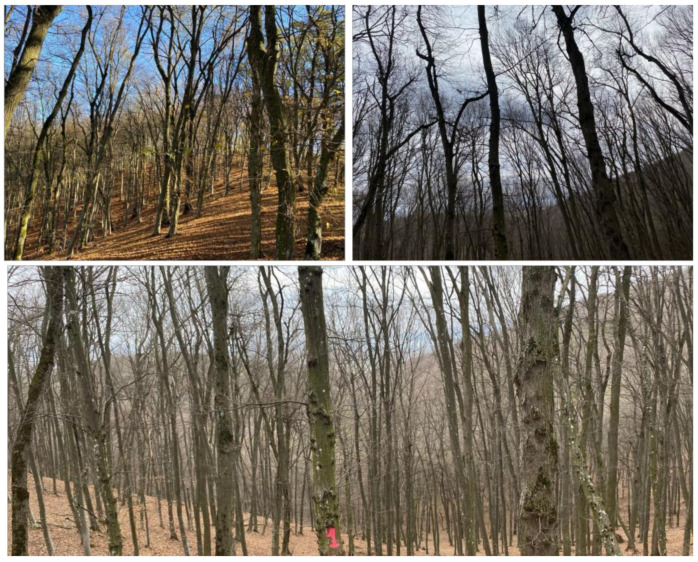
Canopy conditions of the test plot located in a deciduous stand.

**Figure 5 sensors-24-06404-f005:**
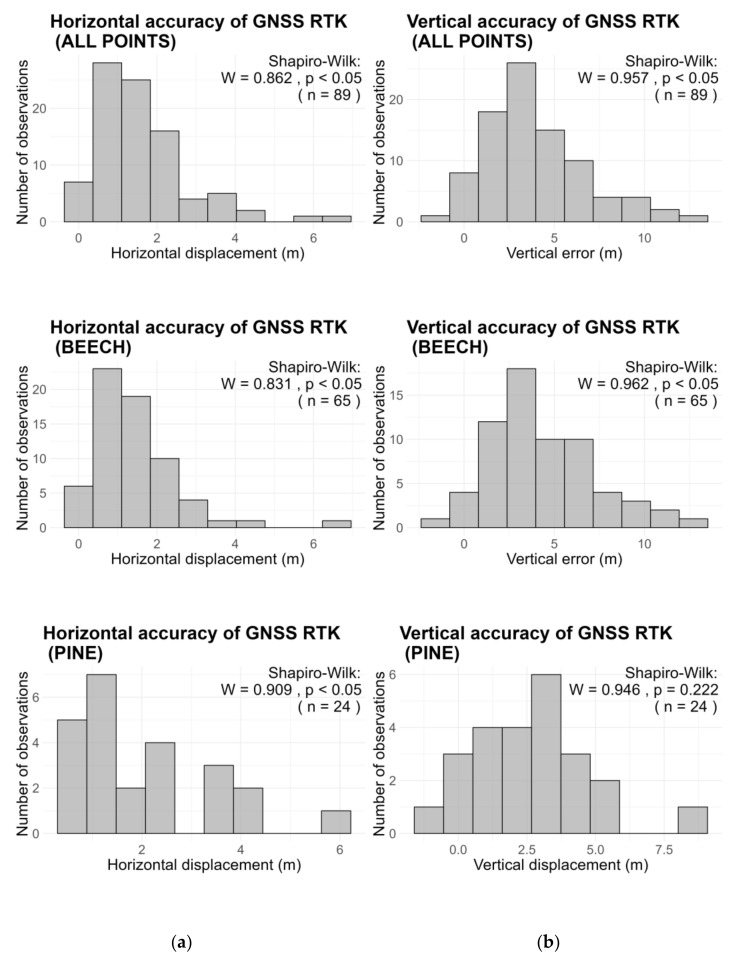
(**a**) Distribution of horizontal errors for GNSS observation in coniferous/deciduous forest cover; (**b**) distribution of vertical errors for GNSS observation in coniferous/deciduous forest cover. In the plot captions: *W*—test statistic for the Shapiro–Wilk normality test, *p*—the level of significance of the *W*-statistic, *n*—the number of samples.

**Figure 6 sensors-24-06404-f006:**
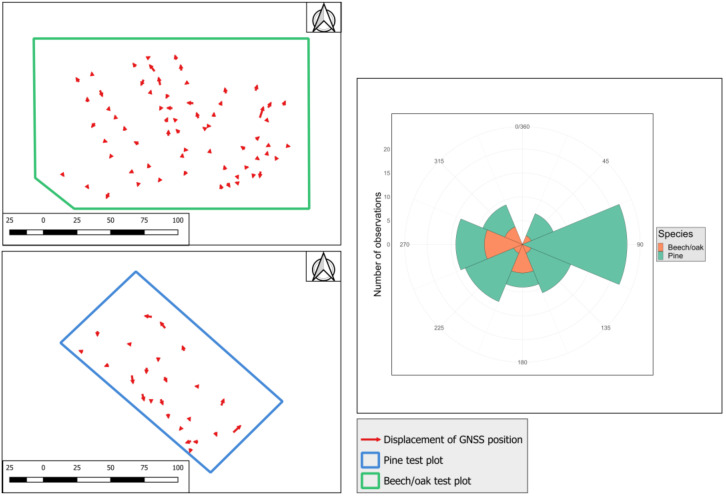
Direction and magnitude of horizontal error (displacement) for GNSS positions.

**Figure 7 sensors-24-06404-f007:**
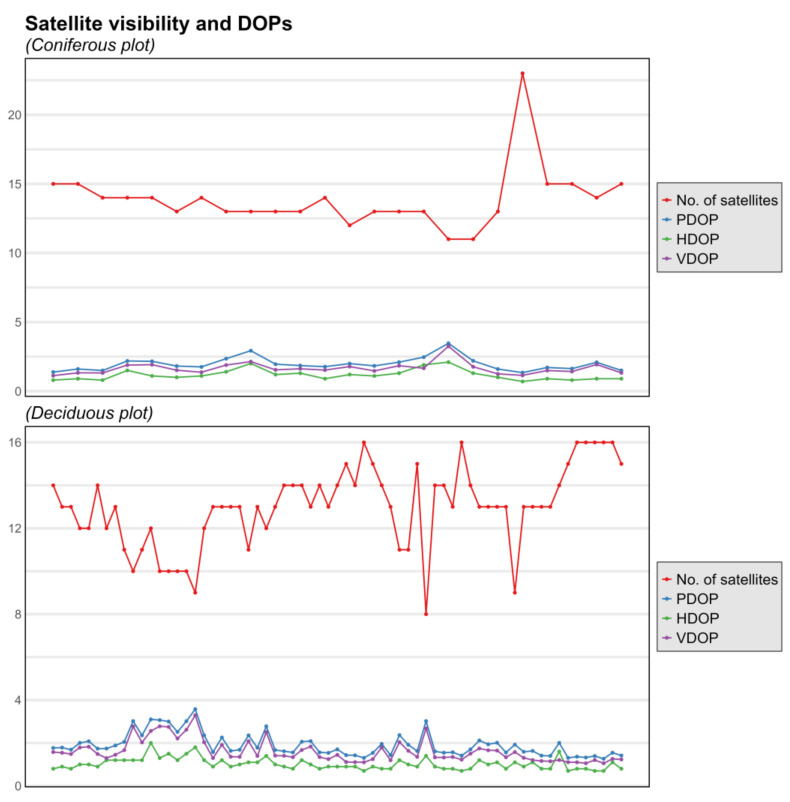
Variation in GNSS-specific conditions for the two test plots.

**Figure 8 sensors-24-06404-f008:**
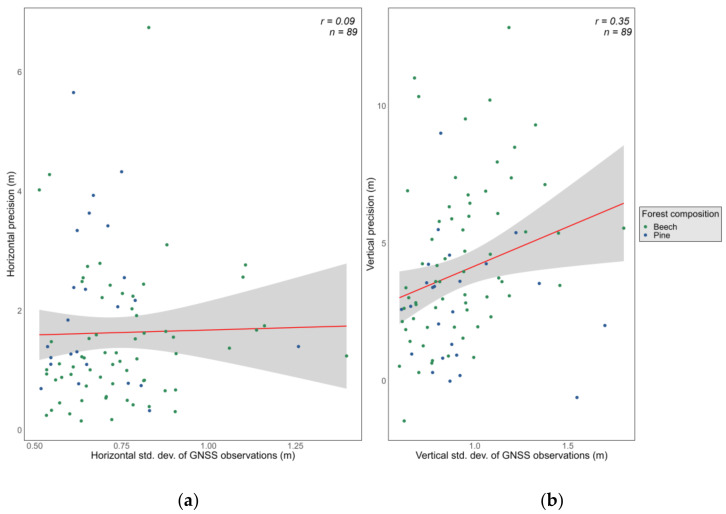
(**a**) Correlation between horizontal accuracy and precision of GNSS-determined positions; (**b**) correlation between vertical accuracy and precision of GNSS-determined positions.

**Table 1 sensors-24-06404-t001:** Forest characteristics of the management units in which the test plots are located.

Test Plot	Avg. Slope (Deg)	Stand Age (Years)	Volume (m^3^/ha)	Canopy Closure (%)	Aspect
Coniferous (pine)	15	105	393	80	S-W
Deciduous (beech/oak)	10	110	169	54	N-NW

**Table 2 sensors-24-06404-t002:** Dual coordinate pairs of control points for closed-loop traverse calculation.

Point no.	X (Easting) (m) ^1^	Y (Northing) (m) ^1^	Z (m) ^2^
1	550,791.879	468,996.465	597.852
	550,791.828	468,996.423	597.771
Diff.	0.051	0.042	0.081
2	550,772.404	550,772.479	605.206
	550,909.363	550,772.404	605.194
Diff.	0.041	0.075	0.012
7 ^3^	550,909.363	468,957.217	617.306
	550,909.353	468,957.209	617.268
Diff.	0.010	0.008	0.038
8	550,985.187	469,003.180	649.158
	550,985.181	469,003.165	649.151
Diff.	0.006	0.015	0.007
14	551,034.388	469,007.828	652.018
	551,034.387	469,007.824	652.005
Diff.	0.001	0.004	0.013

^1^ Coordinates are in the Romanian National Grid System (Stereographic 1970, EPSG code: 31700). ^2^ Heights above MSL (Mean Sea Level), referenced to the Black Sea 1975 System. ^3^ This point is common to the 2 closed-loop traverses.

**Table 3 sensors-24-06404-t003:** Vertical and horizontal accuracies of GNSS observations.

Sample	MAE (m)	Bias (m)	Median Error (m)	Std. Dev. (m)	Min. Error (m)	Max. Error (m)	RMSE (m)
Horizontal accuracy							
All observations (*n* = 89)	1.63	-	1.28	1.21	0.15	6.74	2.03
Pine observations (*n* = 24)	2.07	-	2.07	1.36	0.32	5.66	2.47
Beech/oak observations (*n* = 65)	1.47	-	1.21	1.12	0.15	6.74	1.84
Vertical accuracy							
All observations (*n* = 89)	4.01	3.96	3.47	2.82	−1.46	12.80	4.85
Pine observations (*n* = 24)	4.45	4.40	3.61	2.92	−1.46	12.80	5.27
Beech/oak observations (*n* = 65)	2.81	2.76	2.65	2.18	−0.61	9.00	3.49

**Table 4 sensors-24-06404-t004:** Variation in GNSS-related factors.

Factor	Mean	Median	Std. Dev.	Minimum	Maximum
Sample: all observations (*n* = 89)					
PDOP	1.92	1.77	0.52	1.27	3.58
HDOP	1.06	1.00	0.31	0.70	2.10
VDOP	1.62	1.49	0.49	1.06	3.29
No. of satellites	13.24	13.00	2.01	8	23
Float solutions (out of 30)	0.91	0.00	2.63	0	14
Sample: pine observations (*n* = 24)					
PDOP	1.97	1.84	0.49	1.34	3.46
HDOP	1.17	1.10	0.38	0.70	2.10
VDOP	1.65	1.53	0.44	1.12	3.25
No. of satellites	13.88	13.50	2.25	11	23
Float solutions (out of 30)	1.38	0.00	3.02	0	13
Sample: beech/oak observations (*n* = 65)					
PDOP	1.90	1.70	0.54	1.27	3.58
HDOP	1.02	0.90	0.27	0.70	2.00
VDOP	1.61	1.41	0.51	1.06	3.29
No. of satellites	13.00	13.00	1.87	8	16
Float solutions (out of 30)	0.74	0.00	2.48	0	14

**Table 5 sensors-24-06404-t005:** Distribution of trees around test points.

Radius Considered (Meters)	Avg. no. of Trees	Min. no. of Trees	Max. no. of Trees	Std. Dev. of the no. of Trees
2	0.34	0	4	0.64
4	1.71	0	5	1.21
6	4.19	0	11	2.31
8	7.61	0	16	3.66
10	11.99	1	25	5.74

**Table 6 sensors-24-06404-t006:** Relative importance of factors on GNSS horizontal/vertical accuracy and precision.

Metric	Sample	Proportion of Variation Explained (%)	Most Important Factors ^1^
Horizontal accuracy	All points	25.45	NDVI (19%), Species (11%), No. of trees in a 4 m radius (9%), No. of satellites (8%)
Vertical accuracy	All points	27.40	Slope (22%), Species (19%), No. of satellites (9%), No. of trees in an 8 m radius (9%)
Horizontal accuracy	Coniferous	85.63	No. of trees in a 4 m radius (18%), Distance to nearest tree (17%), Avg. dist. to trees in a 4 m radius (13%), No. of satellites (8%)
Vertical accuracy	Coniferous	85.60	No. of trees in a 10 m radius (18%), Slope (14%), No. of Float solutions (12%), No. of trees in an 8 m radius (10%)
Horizontal accuracy	Deciduous	17.62	No. of satellites (16%), No. of trees in a 10 m radius (15%), No. of trees in an 8 m radius (13%), NDVI (11%)
Vertical accuracy	Deciduous	28.78	Slope (23%), Avg. dist. to trees in a 4 m radius (15%), NDVI (14%), No. of satellites (8%)
Horizontal precision	All points	39.82	PDOP (40%), HDOP (20%), No. of satellites (13%), Species (9%)
Vertical precision	All points	45.41	PDOP (31%), VDOP (29%), No. of satellites (12%), Slope (8%)
Horizontal precision	Coniferous	97.81	No. of satellites (33%), Slope (11%), No. of trees in an 8 m radius (10%), No. of trees in a 6 m radius (7%)
Vertical precision	Coniferous	85.60	No. of trees in a 10 m radius (18%), Slope (14%), No. of Float solutions (12%), No. of trees in an 8 m radius (10%)
Horizontal precision	Deciduous	55.65	PDOP (30%), HDOP (28%), No. of satellites (11%), No. of trees in a 10 m radius (5%)
Vertical precision	Deciduous	28.78	Slope (23%), Avg. dist. to trees in a 4 m radius (15%), NDVI (14%), Distance to nearest tree (7%)

^1^ the number in parentheses after each factor is the contribution of that factor to the total amount of variation explained by all factors, for that specific case.

## Data Availability

Data collected for this study will be made available by the authors upon request.

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
