# Peer review of "Performance Evaluation of Real-Time Kinematic Global Navigation Satellite System with Survey-Grade Receivers and Short Observation Times in Forested Areas"

_sensors, 2024, doi:10.3390/s24196404_

Round 1
Reviewer 1 Report
Comments and Suggestions for Authors
Forestry surveys are an important application of GNSS, where complex obstructions often occur, leading to decreased reliability and accuracy in satellite navigation. This paper provides a detailed assessment of GNSS positioning performance in dense forest environments, which offers valuable reference. However, a few minor issues need clarification:
- During measurements at different reference points, did the receiver maintain continuous tracking of satellite signals, or was it powered off and restarted (cold start) for each observation?
- When analyzing the positioning accuracy, species and slope actually affect the number of satellites and PDOP, which in turn affect positioning accuracy. Please consider the effects holistically rather than treating species and slope as separate factors.
- Please check whether the receiver includes SNR (Signal-to-Noise Ratio) settings, as the quality of observations is affected by the observation environment. This setting is crucial for ensuring positioning accuracy.
Author Response
Please see the attachment for our responses to your observations, thank you.

Reviewer 2 Report
Comments and Suggestions for Authors
The study discusses Performance evaluation of RTK-GNSS and considered survey-grade receivers and short observation times, the work is with innovation, here are my comments:
1. the abstract section did not highlight the innovation of the manuscript and emphasized the accuracy in the horizontal direction (RMSEs of 1.47 – 2.07 meters).
2. line 69-72: The statement “A certain impact of forest conditions on GNSS performance is commonly reported, but rarely with statistically significant correlations established” in the manuscript is too absolute. Does the reference [6,9,15,26,31–33] list provided by the author clearly state this viewpoint.
3. From the introduction in“2.2. Field data collection” part, it can be seen that the manuscript has done a lot of work before selecting effective data. But please check Table 2 to see if the unit in the header is correct. Errors can be marked as decimals, such as 0.01.
4. If the formulas mentioned in the manuscript are cited, please provide additional references.
5. line 273-275: The font in Figure 5 is too small and blurry and explain the parameters such as w, p, n in the diagram, please optimize Figure 5.
6. Please check if all the reference formats in the manuscript are correct.
7.Relatively good accuracies were obtained, especially for horizontal position (RMSEs of 1.47 – 2.07 meters): meter level accuracy seem not accurate?
Author Response

(The authors gave the same response as above.)

Round 2
Reviewer 2 Report
Comments and Suggestions for Authors
The author has response to all my comm. Now can be accepted.